

# A multi-source 120-year U.S. flood database with a unified common format and public access

Zhi Li[1], Mengye Chen[1], Shang Gao[1], Jonathan J. Gourley[2], Tiantian Yang[1], Xinyi Shen[3], Randall Kolar[1], Yang Hong[1]

[1]Hydrology and Water Security Program, Civil Engineering and Environmental Sciences, University of Oklahoma, Norman, 73072, USA
[2]NOAA National Severe Storms Laboratory, Norman, 73072, USA
[3]Department of Civil and Environmental Engineering, University of Connecticut, Storrs, CT, 06269, USA

*Correspondence to*: Yang Hong (yanghong@ou.edu)

**Abstract.** Despite several flood databases available in the U.S., there is a benefit to combine and reconcile these diverse data sources into a comprehensive flood database with a unified common format and easy public access in order to facilitate flood-related research and applications. Typically, floods are reported by specialists or media according to their socioeconomic impacts. Recently, data-driven analysis can reconstruct flood events based on in-situ and/or remote-sensing data. Lately, with the increasing engagement of citizen scientists, there is the potential to enhance flood reporting in near-real-time. The central objective of this study is to integrate information from seven popular multi-sourced flood databases into a comprehensive flood database in the U.S., made readily available to the public in a common data format. Natural Language Processing, geocoding, and harmonizing processing steps are undertaken to facilitate such development. In total, there are 698,507 flood records in the U.S. from 1900 to the present. The database features event locations, durations, date/times, socioeconomic impacts (e.g., fatalities and economic damages), and geographic information (e.g., elevation, slope, contributing area, and land cover types retrieved from ancillary data for given flood locations). Finally, this study utilizes the flood database to analyse flood seasonality within major basins, and socioeconomic impacts over time. It is anticipated that thus far the most comprehensive yet unified database can support a variety of flood-related research, such as a validation resource for hydrologic or hydraulic simulations, hydroclimatic studies concerning spatiotemporal patterns of floods, and flood susceptibility analysis for vulnerable geophysical locations. The dataset is publicly available with the following DOI: https://doi.org/10.5281/zenodo.4547036.

## 1 Introduction

Floods are one of the most common and costliest natural hazards globally (World Health Organisation). In fact, around 74% of natural hazards between 2001 and 2018 were water-related, among which floods were the most devastating. In the United States, eight of the ten costliest weather disasters (in the billions of USD) were floods between 1980 and 2019 (https://www.ncdc.noaa.gov/billions/events), and almost 10% of the flash floods have resulted in agricultural and economic



losses beyond $100,000 (U.S. dollars) per event (Gourley et al., 2017). The flood-producing storms and hurricanes frequently strike the coastal regions with devastating socioeconomic impacts, among which the most damaging Hurricane Katrina affected nine states and resulted in monetary losses over 168 billion USD. Moreover, under the influences of climate change, the increasingly intensified hydrologic cycle and sea level rise pose more threats to coastal areas (Alfieri et al., 2016; Tabari, 2020). IPCC AR5 (2013) has reported that the frequency and intensity of floods in the U.S. are changing, which challenges current water-related infrastructure and water management principles. In light of flood risks, a compilation of a comprehensive flood database can provide insights of both national and regional flood characteristics.

A brief list of data publications on flood disasters is summarized in Table 1 for different regions around the world. Many published works have hitherto been limited to developed countries such as European countries and the U.S. Developing countries either restrict data sharing or lack the resources to collect and assemble flood events. With respect to the available period, not many works continuously offer up-to-date flood data accessible to the public or research communities (e.g., Fiorillo et al., 2018; He et al., 2018; Luu et al., 2019; Petrucci et al., 2019; Shi, 2003). However, it is noteworthy that there are means of collecting flood information. Conventionally, flood reports are produced by local specialists with limited and sometimes delayed information (e.g., Filrilo et al., 2018; He et al., 2018; Luu et al., 2019; Petrucci et al., 2019). Later on, media outlets (e.g., newspaper) start to participate in timely flood reporting, but typically on the high-impact floods (e.g., Hilker et al., 2009; Shi, 2013; Smith et al., 2012; Vos et al., 2010). Insurance companies collectively offer valuable information on flood damages and people affected from a financial perspective (Swiss Re, 2010). Until recently, the increasing engagement of social scientists greatly supports near-real-time flood reporting with web or mobile applications (Chen et al., 2016; de Bruijn et al., 2019), although these reports are often confined to populated, urban areas. In addition to human-led reporting, stream gage and opportunistic sensors (e.g., surveillance cameras, ground radars, and satellites) can also augment flood monitoring in real-time (Hall et al., 2015; Shen et al., 2019).

Despite long-established flood records (reports), there are few studies attempting to merge multi-source flood databases, especially considering the increasing number and diversity of flood databases available. The motivations of a merged dataset are primarily two-fold. First and foremost, we are still under-utilizing all sorts of flood information that can be used for model validation and flood risk analysis (Scotti et al., 2020). Second, each individual dataset has its own limitations, and thereby no single database holistically describes flooding in a given region (Gourley et al., 2013). For instance, flood reports by government agencies or the media are skewed towards high-impact events, whereas local community-level, low-end floods are oftentimes ignored. In light of these motivations, efforts should be undertaken to collectively merge all possible sources to provide off-the-shelf data support to complement flood-related research. Gourley et al. (2013) assembled a georeferenced U.S. database from three primary sources: 1) discharge observations from the U.S. Geological Survey (USGS), 2) flood reports by the National Weather Service from 2006 to 2013, and 3) witness reports from the public. Amponsah et al. (2018) merged a high-resolution flash flood database in Europe with a set of spatial data, rainfall data, and discharge data from 1991 to 2015. Petrucci et al. (2019) collaboratively harmonized five regional flood databases from 1980 to 2015 in the

Mediterranean region to investigate the causes of deaths in flood events. These merged datasets are relatively short in time and
65  not complete. In this study, we introduce a comprehensive United States Flood Database - USFD, which compiles seven
individual databases and converts them into a common data format. Sources to compile this database include 1) reports from
news media, 2) reconstructed flood events from gage and satellite instruments, and 3) crowdsourcing data queried from the
web and mobile applications. As a result, a 120-yr flood database in the U.S. is assembled, unified, and published for public
access, as well as an interactive web interface for immediate use. It is anticipated that this database can support a variety of
70  flood-related research, such as a validation source for hydrologic/hydraulic simulation, climatic studies concerning
spatiotemporal patterns of floods given this long-term and U.S.-wide coverage, and flood risk analysis for vulnerable
geophysical locations. Primary assessments on flood occurrences across the U.S., flood seasonality within major basins, and
socioeconomic impacts across time are carried out to share insights on U.S. floods.

This article is structured as follows. Sect. 2 details seven individual databases and ancillary datasets used to form our
75  database. Sect. 3 describes methods to retrieve (query), clean, and unify these datasets in a processing pipeline. Lastly, Sect 4.
serves as a pre-assessment on floods in the U.S. over the past 120 years, spatially aggregated by geopolitical boundaries and
for major U.S. river basins.

## 2 USFD database components

### 2.1 Individual databases

80  In this section, we detail seven individual databases, which are the NOAA National Weather Service (NWS) storm
reports, Emergency Events Database (EM-DAT), Dartmouth Flood Observatory database (DFO), the University of
Connecticut Flood Events Database (FEDB), cyber-infrastructure flood database (CyberFlood), meteorological Phenomena
identification near the ground data (mPing), and Global Flood Monitoring (GFM).

### 2.1.1 National Weather Service storm reports

85  The NOAA NWS routinely publishes post-event reports of floods from trained spotters, local authorities, and
emergency management officials. This dataset is arguably the most exhaustive meteorology-driven reporting in the U.S. The
descriptors can be mainly categorized into the geophysical location (e.g., begin and end location), time period (e.g., begin and
end time), causes (e.g., heavy rain), impacts (e.g., fatalities and damages), and narratives (see technical documentation for
details: https://www.nws.noaa.gov/directives/sym/pd01016005curr.pdf). Limitations of this database for flood events are
90  summarized in Gourley et al. (2013) such as 1) imprecise event location, 2) times related to meteorological events, 3) relying
on in-person witness accounts, and 4) limited information about the site exposure to antecedent condition. We retrieve all flood
records from 1950 to the present, which totals 144,313 reports.





### 2.1.2 Emergency Events Database (EM-DAT)

The EM-DAT database is produced and maintained by the Centre for Research on the Epidemiology of Disasters
(CRED) in Belgium, which contains all types of global natural disasters in the world from 1900 to the present. These recorded
events should meet one of the following criteria: 1) >10 people dead, 2) >100 people affected, 3) declaration of a state of
emergency, or 4) a call for international assistance. The sources of information stem from government agencies, non-
government organizations, insurance companies, research institutes, and press agencies. The EM-DAT provides information
including geographic location, time entry, fatalities, and economic damages. All the flood-related data entries are collected via
public access at https://www.emdat.be/. Due to its reporting criteria, there are only 189 events recorded in the U.S.

### 2.1.3 Dartmouth Flood Observatory (DFO)

The DFO data, regarded as one of the most popular flood databases in the world, collects flood events from news,
government agencies, and stream gauges and remote-sensing instruments from 1985 to the present (Brakenridge, 2020).
Different from other databases, the DFO collectively retrieves spatial flood information from satellite remote-sensing products,
such as the Moderate Resolution Image Spectroradiometer (MODIS), Sentinel-1, and Landsat. Flood extent is accordingly
provided as shapefiles for easy integration into Geographic Information Systems software. The tabular data includes
geophysical location, date/time, fatalities, affected area, displaced people, flood severity, and primary causes. However, events
without significant river flooding are not included in this database, and they are subject to uncertainties from satellite-derived
flood extent such as water-like echoes in urban areas and limitations due to cloudiness. 469 events have been retrieved from
110 its tabular data in the U.S.

### 2.1.4 University of Connecticut Flood Events Database (FEDB)

Taking advantage of nation-wide flow records at 6,301 stations operated by the U.S. Geological Survey (USGS) and
radar rainfall measurements, a comprehensive flood database is reconstructed from 2002 to 2013, using the characteristic point
method (Shen et al., 2017). At each gauge site, flood events are identified by baseflow separation and filtered with non-
115 significant peaks (i.e., less than 95th percentile). Additionally, flood-producing rainfall events are traced within a certain time
window to portray an event. The FEDB provides shapefiles of stream gauges with a series of flood event attributes (e.g., flow
peak, flow period, rainfall event period, base flow, rainfall-runoff coefficient, and spatial moments-based characteristics). The
original dataset is retrieved from https://ucwater.engr.uconn.edu/fedb. Limitations of this database are the reconstructed events
may not necessarily lead to damages, which may undermine its role in flood impact-related research, and the flood events must
occur in USGS-gauged basins. Over 542,000 events have been reconstructed in the U.S., making this flood database being the
biggest contributor to the combined database.



### 2.1.5 CyberFlood

CyberFlood is a crowdsourced flood database by collecting event reports in a web application developed at the University of Oklahoma (Wan et al., 2014). It is regarded as one of the first integrated systems that collect, organize, visualize,
and manage a flood database globally. We queried the latest results of CyberFlood, which contains flood events, geographic locations, date/time, country code, causes, and fatalities. The latest version of CyberFlood has 203 flood records from 1998 to 2008. To facilitate data unification, we convert all the code-based descriptors to strings (i.e., country and causes) with key matching methods.

### 2.1.6 meteorological Phenomena Identification Near the Ground (mPing)

The mPing app is a crowdsourcing, weather-reporting software jointly developed by NOAA National Severe Storms Laboratory (NSSL) and the University of Oklahoma (Elmore et al., 2014). Members of the public who downloaded this app based on their GPS-enabled smartphones can report the weather event at their locations. Time, geophysical coordinates, standard event types (e.g., flood events classified into four severity levels, tornado, precipitation type, wind). The four flood severity levels are based on the Flash Flood Severity Index (FFSI) proposed by Schroeder et al. (2016). One major limitation
for crowdsourcing data lies in the data validation, as some events are improperly misreported or even hacked. Chen et al. (2013) compared these reports to ground radar observations with respect to precipitation types, and a satisfying correspondence is found between the two. mPing data provides REST API for research-purpose uses, and we queried flood-related events from 2013 to the present with 5000 flood events counted.

### 2.1.7 Global Flood Monitoring (GFM)

The GFM data is produced and managed by de Bruijn et al. (2019), with over 88 million Twitter tweets over the globe since 2014. Contents tied to flood observations are filtered with the Natural Language Processing (NLP) tool BERT, which extracts time of observation and toponyms (in token) and assigns reports to the database attributes after a quality assessment. It is found in the study of Bruijn et al. (2019) that around 90% of the events are correctly detected when compared to another disaster database. Table attributes include event_id, location_id, location_ID_url, country_ID, country_ISO3, and the time of
detection. Due to privacy issues, all the locations are archived in tokens, which requires further decoding. Data is publicly accessible at https://www.globalfloodmonitor.org/download. Given the latest database, we retrieved 6315 flood events in the U.S. and subsequently processed them as described in Sect. 3.

### 2.2 Ancillary datasets

Since one purpose of this database is for flood susceptibility analysis, contributing factors to flooding are also
incorporated for a given location. Land Use Land Cover (LULC), Digital Elevation Model (DEM), slope, distance to a major river, drainage area, and 500-yr flood depth are factored into the data attributes. The LULC value is retrieved from the



Copernicus Global Land Service (CGLS) at 100-m resolution, covering urban, cultivated land, forest, vegetation, wetland, water, and ice. The topographic inputs (i.e., DEM and slope) are acquired from the NASA Shuttle Radar Topography Mission (SRTM) at 90-m spatial resolution, and hydrography dataset (i.e., river networks, drainage area) are acquired from MERIT

Hydro at the same resolution (Yamazaki et al., 2019). The 500-yr flood depth is downloaded from the Joint Research Centre Data Catalogue at https://data.jrc.ec.europa.eu/collection/floods at 1-km spatial resolution. All the extensive computations (i.e., sampling) are processed using the Google Earth Engine platform (Gorelick et al., 2017).

## 3 Processing methods

Figure 1 displays the processing flowchart, including pre-processing, merging, and unifying all seven individual

databases comprising the USFD v1.0. There are 22 descriptors in the database, including the start and end time (UTC), duration (days), longitude and latitude (decimal degrees), toponyms (country, state, and location), flood impacts (i.e., affected area, severity, damage, and fatality), sources (i.e., source database, source ID, collecting sources), event description, and environmental variables as mentioned in Sect. 2.2. Detailed header descriptions and pre-processing steps are summarized in Table 2. Two intrinsic factors describing flood events are date/time and locations. The date/time information varies in different

databases. Some early reports do not record the precise date/time of an event onset, using only year or year and month. For clarity, we format them in a concatenated string with maximum available information. 2015-06-23 00:00:00 is recoded as '20150623000000', and 2015-06 is inserted as '201506'. The date/time of all records is converted to UTC.

It is challenging to completely retrieve the location of events from some databases. The NWS storm report has some missing entries in geographic coordinates, but instead, it has detailed narratives. To compensate, we use the NLP toolkit

provided by the spaCy package, which contains pre-trained models for English multi-tasks. spaCy firstly tokenizes the event narratives and subsequently parses and tags each word with respective entities. Then, we can geocode locations into geographic coordinates via calling the Google Map API. The GFM also does not contain precise geophysical locations to protect user privacy. Therefore, the geographic coordinates are inferred by first converting location tokens into administrative locations (e.g., cities and villages) via the GeoNames API and then geocodes them into geographic coordinates. The state names of a

merged dataset are firstly validated with geophysical locations from an inverse geocoding. If they do not match, a new name from GeoNames is assigned to replace the original one. Meanwhile, the empty fields are also filled during this process. We also processed supplementary flood information such as affected areas and damages from the original database. The affected area for each event is calculated by assuming a circular area whose radius is approximated by the recorded range if available. For economic damages, we sum up all available sub-category damages (e.g., agriculture, property, and structure) to give a

holistic view. For those single databases that do not provide information or information that cannot be inferred from the specific header, we uniformly treat them as Not A Number (NAN) values. As a result, we merged 698,507 total flood records in the U.S. from 1900 to the present. In the DOI link, a merged database USFD, along with seven individual databases, are provided in either comma-separated format or Excel format for general readability. Additionally, an interactive web interface is built





and hosted on http://hydro.ou.edu/research/us-flood-database/, where users can do immediate analysis online and download
the datasets.

## 4 Pre-assessment

### 4.1 Nationwide distributions

Figure 2a shows the nationwide distribution of flood events at state levels. Because the total event numbers might be
skewed by replicated events in different databases, we standardize the total numbers by their maxima to reveal the relative
counts for state-based comparisons. North Carolina, Texas, Missouri, and Pennsylvania are the top-listed regions with over
50% of the total population, among which North Carolina experiences the most cases. From the meteorological perspective,
North Carolina is prone to flooding due to a mixture of a flood-generating mechanism, with landfalling tropical cyclones and
extratropical systems being the primary large-scale drivers, in conjunction with warm-season thunderstorms. In the meanwhile,
the combination of snowmelt and rain-on-snow contributes to flood peak occurring on the lee side of Appalachian Mountains
(Smith et al., 2011). The most devastating flood, estimated as a 500-yr flood, was caused by Hurricane Floyd and led to 35
fatalities. Texas, similar to North Carolina, is affected by tropical cyclones and hurricanes, which produce compound inland
and coastal flooding (Li et al., 2020). Anthropogenic effects such as urbanization and regulation, apart from meteorological
effects, are equally critical for inland flooding. Blum et al. (2020) in a recent study noted that these listed regions experienced
increased urbanization from 1974 to 2012, resulting in an average 3.3% increased flood magnitude by changes in impervious
cover alone.

Figure 2b lists detailed event numbers for each state and the composition of each individual flood database. The
FEDB contributes a major portion of the unified database because of the data length of flow records, and states with higher
gauge densities undoubtedly yield more event numbers than gauge-sparse regions. The data nonuniformity underlies a major
limitation of this specific dataset. Regions with more exposure to observational sources (e.g., densely populated and gauge-
205 dense areas) likely have more recorded events. However, it is expected that by including more observations from remote-
sensing sources, this gap can be potentially compensated. Following the composition, NWS storm reports comprise the second
largest number of events because of the long data length (i.e., 70 years). Other databases, such as EM-DAT, though the longest
available length, only records very high-impact events, and the crowdsourcing databases are limited by their short lengths.

### 4.2 Flood seasonality in major water basins

Flood variability is highly associated with seasonal atmospheric pathways of moisture delivery and basin attributes
(Dickinson et al., 2019). In this regard, we segregate the nation-wide events into major basins and months. The Hydrologic
Unit Code (HUC) 4-digit basins, as shown in Fig. 3, are obtained from the national hydrography dataset. Figure 3a depicts
months with the highest number of events. Flooding generally happens between January and June over the majority of the U.S.



basins, similar to that of other studies (e.g., Brunner et al., 2020; Dickinson et al., 2019; Villarini, 2016). The basins are
215 clustered into several regions according to local hydroclimatologies. In the West Coast (e.g., western Washington, Oregon,
and California), flood events are dominant in winter months because of Atmospheric Rivers (AR) as a main driving factor,
which is a carrier of water vapor from the tropics (Ralph et al., 2006). Moving to the East, floods in the Rockies (i.e., Upper
Colorado and Great Basins) are featured by spring snowmelt in snow-fed rivers, whereas in the Desert Southwest (e.g., Lower
Colorado and Rio Grande regions), floods likely occur in late summer, which is ascribed to the North American monsoon and
220 North Pacific tropical cyclones. Closer to the Gulf of Mexico, flooding events during late spring and summer are due to severe
thunderstorm activity and mesoscale convective systems. The lower Mississippi, Ohio, and Tennessee river basins experience
their biggest floods during the spring from extratropical cyclones (Lavers and Villarini, 2013). The lower Florida Peninsula
features high numbers of summer flood events, which are tied to North Atlantic tropical cyclones (Villarini et al., 2014). In
the northeastern U.S., tropical cyclones, winter-spring extratropical cyclones, warm-season thunderstorms are the primary
flood agents, yet winter-spring extratropical cyclones account for larger fractions, similar to the study of Smith et al. (2011).
Figure 3b displays the number of flood events within each basin, grouped by months. The Mid-Atlantic region – HUC2 02 –
takes seven places out of the top twenty basins, with the Delaware river basin near the coast (HUC2-0204) being the highest
one. In terms of flood seasonality, it is relatively evenly spanned across seasons and months for these listed basins, which
suggests its susceptibility to widespread floods. This symmetric feature around the Appalachian Mountains across seasons is
230 also highlighted in Villarini (2016), in which they suggest flow regulations play an essential role in weakening the seasonal
cycle. In a recent study by Brunner et al. (2020), these basins are identified as severe or moderate widespread flooding in space,
and our results indicate these regions also have widespread flooding in time (month).

Figure 4 depicts the flood seasonality for two separate periods, 2000–2010 (a) and 2010–2020 (b), which investigates
potential shifts in flood timing for U.S. river basins. In the West Coast (California and Columbia basin), there is a shift from
235 early-winter flooding to late-winter or early-spring flooding, especially near the northern coast. This probably relates to
snowmelt occurring in early spring, in conjunction with the rain-on-snow effect. The Great Plains feature an earlier maximum
flood frequency, transitioning from early summer to late spring, which relates to enhanced and earlier timing of thunderstorm
activities due to spring warming. The South Atlantic Coast shows a delayed maximum flood frequency from winter to spring.
The lower Florida Peninsula, however, does not present a clear monthly shift, which is still controlled by tropical cyclones.

## 4.3 Flood impact assessment

In the USFD, flood impacts are based on affected areas, economic damages, and fatalities. Since affected areas are
relatively subjective, they are not analyzed in this study. All the economic damages (US dollars) are adjusted for inflation with
GDP deflectors obtained from the World Bank. Figure 5 depicts the fatalities and damages by year. Although events
continuously span from 1900 to the present, impact assessments were not provided in the earlier years (before 1980). The 10-
245 yr running mean generally represents the long-term trend. Both fatalities and damages begin at high rates in the early years,



possibly due to the immature understanding of floods and lack of flood protection measures. The 1964 flood event, known as the "Thousand Year Flood", caused hundreds of millions of damages, and over ten people lost their lives. Since 1990, however, with the improved measures in flood prediction, management, and protection, fatalities rates start to decrease except for some highlighted major events. Yet, in recent years, damages have a slight upward trend, which is tied to frequent floods caused by

250 intensified active hurricane events and anthropogenic effects. For instance, the 2011 and 2012 Atlantic hurricane season are deemed as the third- and fourth-most active hurricane season on record. The 2017 hurricane season featuring Harvey, Irma, and Maria, was the costliest hurricane season on record, which is reflected in flood-related damages. Land surface changes such as urbanization continue to develop a conducive environment for urban flooding.

State-specific damages across time shown in Fig. 6 reveal the trends in flooding hotspots. For Texas and Louisiana,

consistent upward trends are present because of intensified extreme events in the Gulf Coast. The slopes in flood damages are the greatest among the identified hotspots, manifesting severe flood risks. North Carolina has experienced an increase in damages in the past ten years, accompanied by major events during the 2016 and 2018 Atlantic hurricane seasons. Florida, similar to North Carolina, has encountered active hurricane seasons and accompanying damages. In summary, these flood hotspots with increased flood damages should raise awareness from policy-makers and the public.

**5 Conclusions and outlook**

This work presents a merged United States Flood Database (USFD) that features the longest and most comprehensive recording of flooding across the country. The merged database, integrated from multiple sources, can overcome limitations inherent to the individual databases and thus maximize benefits. It is expected that this database can support a variety of flood-related research, such as a validation resource for hydrologic and hydraulic model simulations, hydroclimatic studies

concerning spatiotemporal patterns of floods, and flood susceptibility analysis for vulnerable geophysical locations.

We showcase three analyses based on the developed flood database. For flood occurrences across the U.S., Texas, Pennsylvania, and Missouri are highlighted with great exposures to floods in total amount, which could raise awareness from policy-makers and the public. Flood seasonality in major river basins generally follows the large-scale synoptic weather patterns. In addition, delayed timings of maximum flood frequency are observed in the West Coast and Atlantic River Basin,

possibly due to earlier snowmelt than in prior decades that now contributes to spring floods. Floods in the Great Plains, on the contrary, feature an earlier month of maximum flood frequency, which is possibly tied to intensified thunderstorm activities because of earlier spring warming. Lastly, flood impacts are assessed in terms of economic damages and fatalities, and we found a slight increasing trend in damages in recent years. Especially in Texas and Louisiana, a consistent increase in damages is evident, which relates to intensified storm activity and expanding urban zones. Under a warming climate, storms are

projected to occur more frequently in the future, which challenges current water infrastructures and water management principles.

Notwithstanding, there are some limitations associated with the current version of USFD. First, the individual databases disproportionally make up the merged one. FEDB, taken from streamflow records over a long history, consists of the majority of the flood events. The emerging crowdsourced databases are expected to play a significant role with the increasing engagement of citizen scientists. Space-based observations could markedly bridge the gaps between well-observed urban areas and gauged basins to gauge-sparse areas in rural zones. For instance, complete use of the MODIS imagery onboard Terra and Aqua satellites, in association with Landsat or SAR data, can reconstruct global flood events at daily resolution. In addition to flood extent, flood depth can also be approximated through the use of high-resolution DEM data. Floods, reported by insurance companies, offers another angle to not only record events, but relate flood hazards to societal impacts comprehensively. In the future, we hope to incorporate such a dataset to enrich our database at a global view. Second, the data processing framework needs to be automated in real-time. We plan to migrate processing codes to the cloud, so that they can query records from the child databases and update the parent database on a regular basis.

## 6 Data and code availability

The USFD open-access dataset and all individual datasets are available at https://doi.org/10.5281/zenodo.4547036 (Li et al., 2020b). The Python codes to process, merge, and analyze are publicly accessible at https://github.com/chrimerss/USFD.

## 7 Author contribution

Zhi Li: Conceptualization, Data curation, Formal analysis, Investigation, Methodology, Writing – original draft preparation

Mengye Chen: Investigation, Methodology, Writing – review & editing

Shang Gao: Methodology, Writing – review & editing

Jonathan J. Gourley: Conceptualization, Writing – review & editing

Tiantian Yang: Writing – review & editing

Xinyi Shen: Writing – review & editing

Randall Kolar: Writing – review & editing

Yang Hong: Conceptualization, Investigation, Writing – review & editing, Supervision

## 8 Acknowledgements

The first author is sponsored by the Hydrology and Water Security program and the Hoving fellowship at the University of Oklahoma. We would like to acknowledge the efforts of collecting information from the seven open-source flood databases.



## 9 Competing interests

The authors declare that they have no conflict of interest.



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





**Figure 1: Flow chart of dataset processing.**

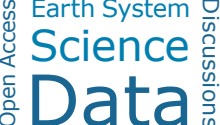

**(a)**

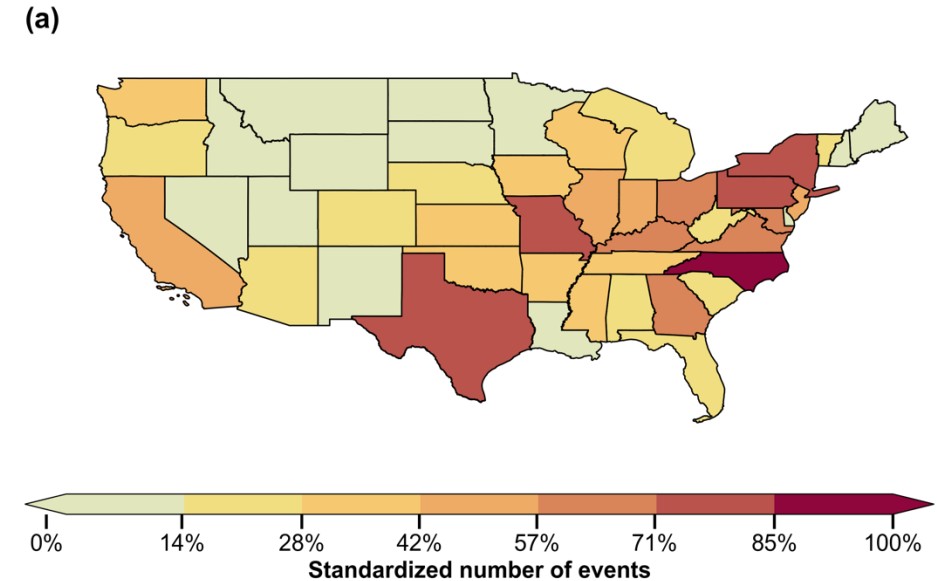

**(b)**

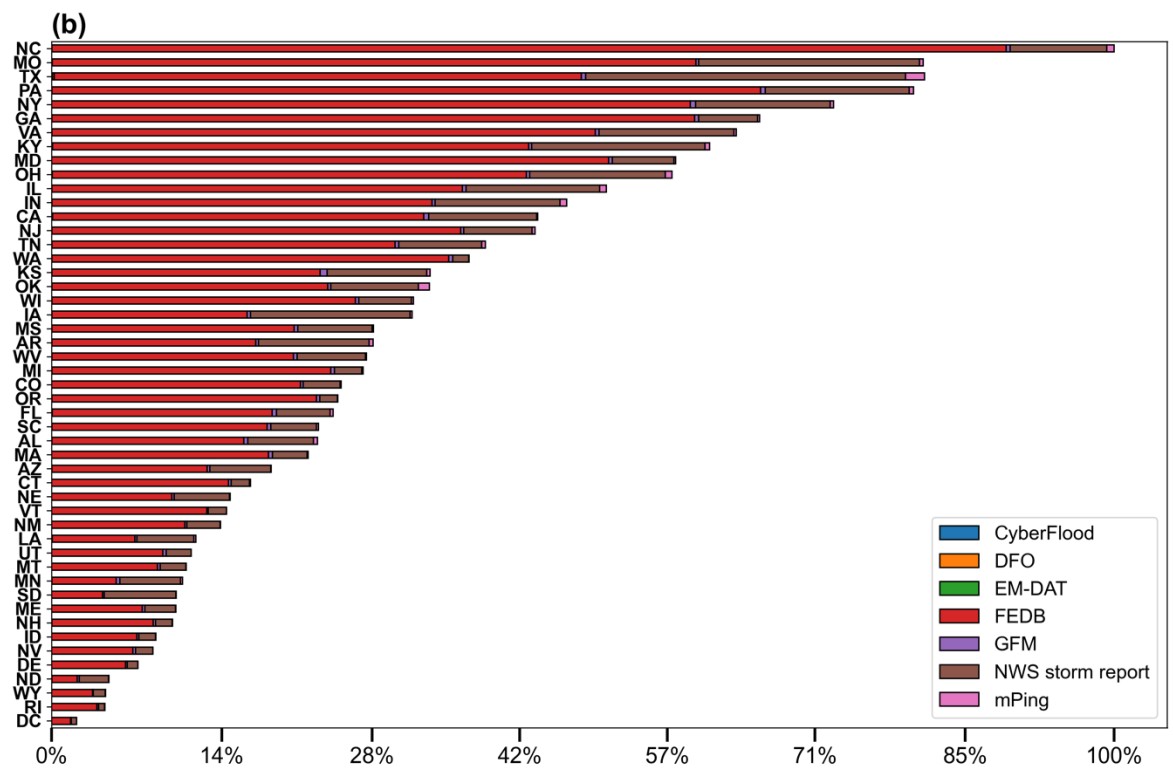





Figure 2. Map of min-max standardized flood occurrences in the U.S. (a) and compositions with respect to different databases.

**(a)**

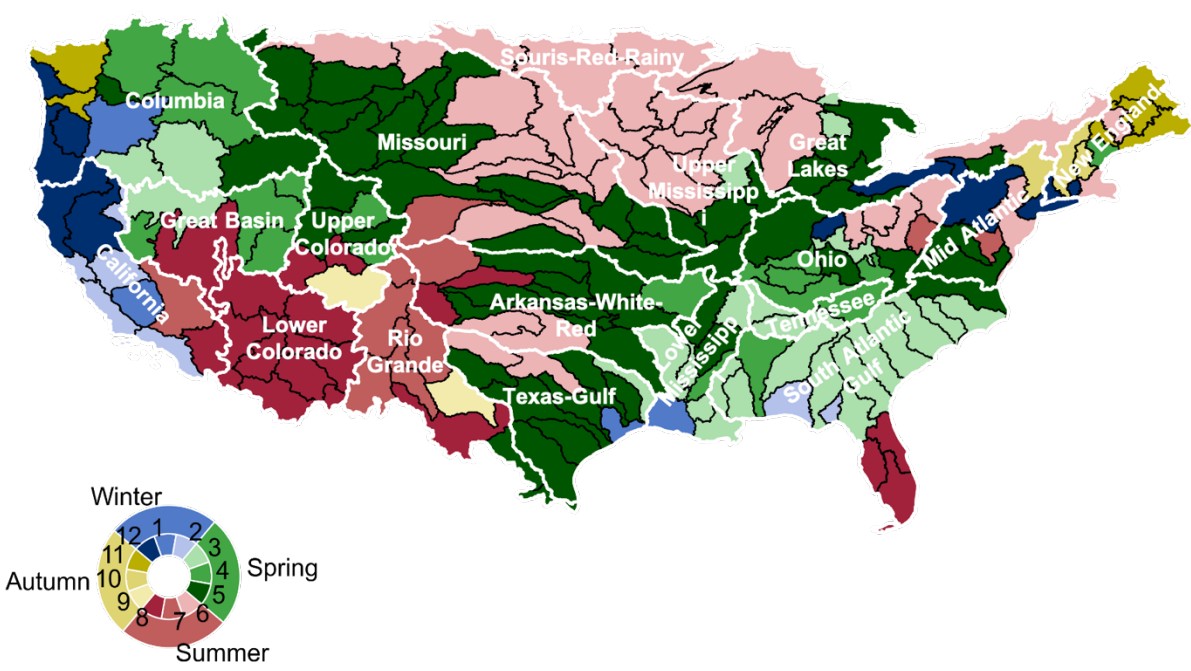

**(b)**

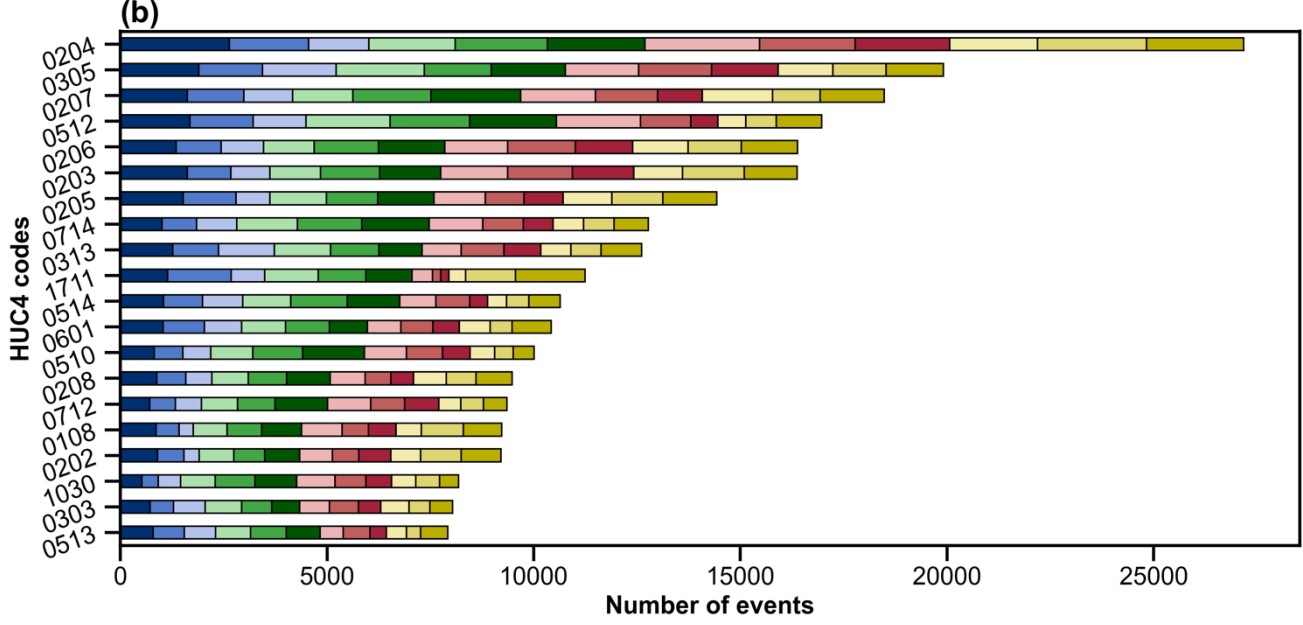



Figure 3. (a) Map of months with largest flood occurrence in major U.S. watersheds (Hydrologic Unit Code 4). (b) flood occurrences of the top 20 basins, goruped by months.

**(a)**

**(b)**

Figure 4. Maps of month of maximum flood frequency, same as Fig. 3a but for two periods: (a) 2000–2010; (b) 2010–2020.

Figure 5. Time series of flood impacts: (a) mean fatalities per year; (b) mean economic damages per year (US dollars based on 2020). The highlighted bars represent 1964, 1997, 2011, 2012, 2016, and 2017, which are the active hurricane seasons on record.

Figure 6. Time series of regional economic damages at 2020 values.





Table 1. A list of databases in literature (grouped by regions and follow a chronological order referring to publication date).



| Name | Available Period | Region | Source | features | Reference |
|---|---|---|---|---|---|
| National Flood Database | 1949 - 2000 | China | Provincial newspaper | Location, datetime, Fatalities, missing people, affected population, direct damages | Shi (2003) |
| Flash Flood Disasters | 2011 - 2015 | China | Official reports | Location, datetime, Fatalities, severity, affected population | He et al. (2018) |
| Swiss flood and landslide damage database | 1972 - 2007 | Switzerland | Press articles | Location, datetime, Damages, fatalities, and injuries | Hilker et al. (2009) |
| MEFF | 1980 - 2015 | Mediterranean | Post flood surveys, newspaper, official reports, media website | Location, datetime, Fatalities and detailed circumstances, causes of deaths | Petrucci et al. (2019) |
| European Flood Database | 1900 - 2015 | Europe | ~7000 hydrometric stations | Daily mean discharge, maximum peak discharge | Hall et al. (2015) |
| EuroMedeFF | 1991 - 2015 | Europe and Mediterranean | hydrometric stations | Location, datetime, rainfall rates, DEM, flood magnitude | Amponsah et al. (2018) |
| Unified Flash Flood Database | 2006 - 2013 | United States of America | Hydrometric stations, official reports, witness reports | Location, datetime, Geographic impacted regions, flow information, fatalities, damages | Gourley et al. (2013) |
| mPING | 2014 - present | United States of America | Citizen scientists | Location, datetime, Flood descriptions | Chen et al. (2016) |
| Flood Events Database | 2002 - 2013 | United States of America | ~6301 hydrometric stations | Peak flow, flow volume, base flow, runoff coefficient, travel distance, | Shen et al. (2017) |



| | | | | covariance of precipitation and travel distance | |
|---|---|---|---|---|---|
| NWS storm report | 1950 - present | United States of America | Officials, specialists, local reports | Location, datetime, Flood causes, fatalities, damages, agricultural damages, event narrative | Storm Events Database (2020) |
| ANADIA DB | 1998 - 2015 | Niger | Local officials | Location, datetime, Settlements involved, people affected, fatalities, crop and livestock losses, infrastructure affected | Fiorillo et al. (2018) |
| National Disaster database | 1989 - 2015 | Vietnam | Official report | Location, datetime, Fatalities, economic losses, housing damaged, education impact, agriculture impact, irrigation impact, transportation impact, fisheries impact, electricity impact | Luu, et al. (2019) |
| Dartmouth Flood Observatory database | 1985 - present | Globe | Media press, Earth Observations, report | Location, datetime, Fatalities, area affected, people displaced, severity, cause | Brakenridge (2020) |
| EM-DAT | 1900 - present | Globe | Media press and reports | Location, datetime, Economic losses, fatalities, people injured, area affected | Vos et al. (2010) |
| Sigma | 1970 - present | Globe | Insurance claims | Location, datetime, Economic losses, fatalities | Swiss Re (2010) |
| CyberFlood | 1998 - 2014 | Globe | Citizen scientists (crowdsourcing) | Location, datetime, severity, fatality, area affected | Wan et al. (2014) |
| Global Flood Monitor | 2015 - present | Globe | Citizen scientists (Twitter) | Event location and datetime | de Bruijn et al. (2019) |





Table 2. Descriptors in the USFD database.

| Headers | Unit/format | preprocessing | description | Sample |
|---|---|---|---|---|
| DATE_BEGIN | yyyymmdd(hh:mm:ss) | Pattern extraction, converting | Event beginning datetime (UTC) | 202011042249 |
| DATE_END | yyyymmdd(hh:mm:ss) | | Event end datetime (UTC) | |
| DURATION | days | End date – begin date | Duration of an event | 10 |
| LON | Decimal degree | Check validity; remove unreasonable (lon,lat) | Longitude of an event | (-120,60) |
| LAT | Decimal degree | | Latitude of an event | |
| COUNTRY | String | Mapping country code to full name | Country name | The United States of America |
| STATE | String | Replace or fill names with (lon, lat) | State name | Oklahoma |
| LOCATION | String | None | Location of an event | Bryan |
| AREA | Square kilometers | Calculate affected areas from reported storm range | Event-affected area | 1000 |
| FATALITY | Int | Check data type | Number of fatalities | 43 |
| DAMAGE | US dollars | Sum up sub-category costs; check data type; convert to dollars | Economic damages (direct) | 1e7 |
| SEVERITY | NA | None | Severity of an event (according to Dartmouth Flood Observatory data) | 1,1.5,2 |
| SOURCE | String | None | Collecting sources | Newspaper |



| SOURCE_DB | String | None | The original recorded database | NOAA storm report |
|---|---|---|---|---|
| SOURCE_ID | String | None | The original event ID in source database | 102300 |
| CAUSE | String | None | Causes of a flood event | Heavy rain |
| DESCRIPTION | String | None | Event narratives | River overflowing/bankfull |
| DEM | meters | Retrieved from Shuttle Radar Topography Mission | elevation | 120 |
| SLOPE | degree | Derived from DEM | slope | 10 |
| LULC | class | Retrieved from Copernicus global landcover 2019 | Land use land cover classification | Urban |
| DISTANCE_RIVER | km | Distance computed in Google Earth Engine | Distance from event location to nearest major river | 3.5 |
| CONT_AREA | sqkm | Retrieved from MERIT Hydro | Contributing area | 1.35 |
| 500yr_DEPTH | meters | Retrieved from 500-yr flood depth | 500-yr flood depth | 1.23 |