# Peer review of "A multi-source 120-year U.S. flood database with a unified common format and public access"

_Earth System Science Data, 2021_

## Author Comment (AC1)

Minor revisions in the pre-assessment section are needed to provide more clarity on the analyses performed and the connection between text and figures. For example, the method used to account for replicated events should be more fully explained - it is not clear what the percentages used in Figure 2 represent. Specific comments are provided below, first for providing more clarity and second for figures and tables.

The authors would like to express our appreciation of your valuable comments towards improving our manuscript. Your effort has been acknowledged in our revised manuscript.

Comments:

1. Make it clear in the beginning whether this dataset includes both river and coastal flooding. Highlight that this dataset is the "longest and most comprehensive recording of flooding across the country" in the abstract and introduction like you have mentioned in the conclusions.

Response:
Thanks for your comments. This dataset does include both riverine flooding and coastal flooding together. We have re-emphasized this point in our revised manuscript.

L.70 - "This dataset includes diverse flood subtypes, including riverine flooding, coastal flooding, flash flooding, etc. and features the longest and most comprehensive recording of flooding across the country."

2. Section 2: Are there other commonly used flood datasets which you chose not to include? If so, perhaps mention your dataset selection method.

Response:
Thanks for your comments. We have tried to include all flood databases that are available to support flood-related research, but some datasets produced by individuals or untrustful sources are not selected. We have added the criteria into our text:

L.86 - "Each candidate of this compiled database has to satisfy the criteria: (1) published by trustworthy organizations, (2) has been used in at least one trackable high-impact publications, and (3) contains useful information for flood-related researches"

3. Line 188 "Because the total event numbers might be skewed by replicated events in different databases, we standardize the total numbers by their maxima to reveal the relative counts for state-based comparisons." –unclear what the standardization is and how it accounts for replicated events. Is the maxima the size of the largest set of repeated events? How do you discern replicated events from the dataset?

Response:
Thanks for your comments. We did not intend to discard replicated events because the repeated events in each candidate are related to different attributes and uncertainties, which is up to users to select the one fits into their scope. For visualization, we present the percentage instead of event counts to inspect state-wise comparison, as the total numbers contain repeated events and

thus less informative. The maxima refer to the state having the largest number of events (PA in this case). We refined the text to

L.201 - "Because the total event numbers might be skewed by replicated events in different databases, we standardize the event counts of each state by the maximum number of events, including repetitive events from different sources, out of all states to reveal the relative composition. It is noteworthy that we do not intend to rule out repetitive events because they are reported with different attributes and uncertainties in different candidate databases, which is up to users to select the one that fits into their scopes."

4. Line 226 "The Mid-Atlantic region – HUC3 02 – takes seven places out of the top twenty basins, with the Delaware river basin near the coast (HUC2-0204) being the highest one" – unclear what 'top' and 'highest' are referring to. If you are unfamiliar with HUC 2 and 4, it is difficult to connect this sentence to Figure 3. Bolding the '02' portion of the HUC4 codes in Figure 3 could help, but ultimately it would be very helpful to have a mapping from codes to basin names.

Response:
Thanks for your suggestions. The text has been refined.

L.245 - "The Mid-Atlantic region – HUC2 02 – takes seven places out of the top twenty HUC4 basins listed in Figure 3b, with the Delaware river basin near the coast (HUC2-0204) being the highest one."

We have made corresponding changes to the Figure 3b with HUC2 codes being bolded, and added basin names next to the bars.

[Figure]

Figure 3b. flood occurrences of the top 20 HUC4 basins (HUC2 codes in bold), grouped by months.

5. Line 225 – this paragraph contains a number of claims which need citations.

Response:
Thanks for your comments. We have cited two additional works to support this statement.

L. 241 - "In the northeastern U.S., tropical cyclones, winter-spring extratropical cyclones, warm-season thunderstorms are the primary flood agents, yet winter-spring extratropical cyclones account for larger fractions (Smith et al., 2011; Villarini, 2016)"

Smith, J. A., Villarini, G., & Baeck, M. L.: Mixture Distributions and the Hydroclimatology of Extreme Rainfall and Flooding in the Eastern United States, Journal of Hydrometeorology, 12(2), 294-309, doi: 10.1175/2010JHM1242.1, 2011.

Villarini, G.: On the seasonality of flooding across the continental United States, Adv. Water Resour., 87, 80-91, doi:10.1016/j.adv.watres.2015.11.009, 2016.

6. Line 247 "Thousand Year Flood" – be more specific about what this event was and where it happened.

Response:
Thanks for your comments. We have included additional information in text:

L. 271 - "The 1964 flood event which happened in the Pacific Northwest and northern California during the Christmas holiday, also known as the "Thousand Year Flood", caused hundreds of millions of damages, and over ten people lost their lives."

7. Line 245 "generally" – remove vague language.

Response:
Thanks for your comments, it has been modified.

8. Line 255 – "The slopes in flood damages are the greatest among the identified hotspots, manifesting severe flood risks." - what is the slope referring to - the slope of a hillside at these locations or the slope of the 10-year average plot? How is the slope manifesting flood risk?

Response:
Thanks for your comments. The slope refers to the flood economic damage (US dollars) change per ten years from which a higher slope indicates more cost with time. The increasing cost manifests potential flood risks. We refined the text as follows:

L.281 - "The slopes of annual flood damage curve in the two regions are the greatest among the identified hotspots, manifesting potential flood risks."

9. Line 275 -  need citation for last sentence

Response:

Thanks for your comments, we cited the IPCC report in our text.

IPCC, 2014: Climate Change 2014: Synthesis Report. Contribution of Working Groups I, II and III to the Fifth Assessment Report of the Intergovernmental Panel on Climate Change [Core Writing Team, R.K. Pachauri and L.A. Meyer (eds.)]. IPCC, Geneva, Switzerland, 151 pp.

Comments on figures and tables:

1. Line 38 and Table 1 – discussing data availability throughout the world and the different types of flood reports is valuable context, but providing a table of specific international datasets seems unnecessary as the paper is ultimately about US flood data and the listed international datasets are not used. Consider removing the table.

Response:
Thanks for your suggestions, we have deleted this table.

2. Figure 3 – provide a mapping from the HUC codes to the basin names used in the map.

Thanks for your suggestions, Figure 3 has been modified with HUC codes and names included.

[Figure]

[Figure]

Figure 3. (a) Map of months with largest flood occurrence in major U.S. watersheds (Hydrologic Unit Code 4). (b) flood occurrences of the top 20 HUC4 basins (HUC2 codes in bold), grouped by months.

3. Figure 5 – why does the 10-year average for the fatality plot drop to zero at 2020?

Response:
The mean fatality is measured by the fatalities per event, so it does not reflect the total fatality in 2020. The reason we chose mean fatality is to avoid repeated counts of fatalities in repetitive events. We added this to our text:

L.267 – "To avoid repeated counts of damages or fatalities due to repetitive events, we herein calculate the mean values per event."

4. Figure 3 and 4 -  the wheel legend for coloring by month has increasing value in colors throughout each season except winter. It would be helpful to have the increasing value pattern consistent in all seasons.

Response:

Thanks for your suggestions, the two figures have been updated, as one is shown in comments #2 and the other one is shown below.

[Figure]

Figure 4. Maps of month of maximum flood frequency, same as Fig. 3a but for two periods: (a) 2000–2010; (b) 2010–2020.

---

## Author Comment (AC2)

Overall: Good effort, good compilation, good description, valuable to many users.

The authors would like to express our appreciation of your valuable comments towards improving our manuscript. Your effort has been acknowledged in our revised manuscript.

Comments:

1. Colors of the source databases do not show up in Fig 1b, partly due to color selection and partly due to very low numbers from several sources. Consider better ways to convey this info?

Response:
Thanks for your comments. It is due to the dominance of one or two databases (i.e., FEDB and NWS reports). In order to make all discernable, we applied log transform to event numbers and then replotted as below.

[Figure]

Figure 2 (b) fractions of logarithmic event numbers of each candidate database to logarithmic total event numbers.

2. Somewhere, perhaps a table in an appendix, match the HUC4 individual codes to named river basins or flood regions?

Response:
Thanks for your comments. We re-generated the Figure 3 as follows. HUC4 basin names are shown next to the bars.

[Figure]

Figure 3. (a) Map of months with largest flood occurrence in major U.S. watersheds (Hydrologic Unit Code 4). (b) flood occurrences of the top 20 HUC4 basins (HUC2 codes in bold), grouped by months.

3. Absence of any uncertainty estimates represents a substantial limitation. Authors focus, quantitatively, on numbers of events, but virtually every parameter - including location – listed in Table 2 exists within a substantial uncertainty range. Need expanded discussion of uncertainties! Which could lead to recommendations for reducing said uncertainties (e.g., formal record-keeping, systematic enhanced remote sensing, new guidance for citizen observors). Perhaps add an uncertainty column to Table 2? Even on log scale, bars and trendlines in e.g., Figure 6 have substantial uncertainty? Most users listed by authors will want or need uncertainty information. How to reduce uncertainty going forward to better quantify trends?

Response:

Thank you for your comments and suggestions. For each candidate database, we complete uncertainty discussions such as geographic location, flood reporting, etc. in Section 2. However, it is difficult at this stage to assign data uncertainty levels to our merged product. First, it is subjective to decide which report weighs more over the others. For instance, should we trust more on witness reports or gage readings? Witness reports have uncertainties in reporting the geographical coordinates due to privacy issues, such as the Twitter data being discussed in main text. In addition, some falsely identified reports possibly exist because of internet hacking. Stream gage, although managed by human kinds, is likely to malfunction during extreme flooding events such as Hurricane Harvey. Therefore, we think the appropriate way here is to discuss more uncertainties of our database instead of assigning uncertainties quantitatively or categorically. Beyond that, we put forward recommendations that each flood database at its development stage, should consider or include uncertainty information for users. Therefore, it is more reasonable for a compiled database to also include that information. As a part of future work, we will consider ways of incorporating uncertainty measures from individual database and harmonize them into one.

L.314 - "Second, the uncertainties exist in each candidate database. The crowdsourcing dataset such as web-based may be less reliable because of lack of stringent data scrutiny, especially for studies that require precise flood locations. Developing new guidance for citizen scientists is a remedy. The instrument uncertainties are subject to confined locations such as stream gauges and technical algorithms to retrieve flood information such as remotely sensed observations. Algorithm developers should take into account how to quantify uncertainties in addition to the end product. The flood reports from government agencies are relatively less uncertain. Therefore, it is highly recommended that users select the candidate databases that fit to their research scope. We also encourage each flood database, during its original development, produce uncertainty measures quantitatively. In a future work, we will consider ways of incorporating uncertainty measures from individual flood database and harmonize them into one for delivering comprehensive flood information."